# Seroprevalence of SARS-CoV-2 in Client-Owned Cats from Portugal

**DOI:** 10.3390/vetsci9070363

**Published:** 2022-07-16

**Authors:** Andreia Oliveira, Maria Aires Pereira, Teresa Letra Mateus, João R. Mesquita, Helena Vala

**Affiliations:** 1Escola Superior Agrária de Ponte de Lima, Instituto Politécnico de Viana do Castelo, 4990-706 Ponte de Lima, Portugal; andreia.oliveira@ipvc.pt; 2Hospital Veterinário de Gaia, 4415-369 Pedroso, Portugal; 3Instituto Politécnico de Viseu, Escola Superior Agrária de Viseu, 3500-606 Viseu, Portugal; hvala@esav.ipv.pt; 4Global Health and Tropical Medicine (GHTM), Instituto de Higiene e Medicina Tropical (IHMT), Universidade Nova de Lisboa (UNL), 1349-008 Lisboa, Portugal; 5CERNAS-IPV Research Centre, Polytechnic Institute of Viseu, Campus Politécnico, 3504-510 Viseu, Portugal; 6CISAS-Center for Research and Development in Agrifood Systems and Sustainability, Escola Superior Agrária, Instituto Politécnico de Viana do Castelo, 4900-347 Viana do Castelo, Portugal; tlmateus@esa.ipvc.pt; 7EpiUnit-Instituto de Saúde Pública da Universidade do Porto, Laboratory for Integrative and Translational Research in Population Health (ITR), 4050-091 Porto, Portugal; jrmesquita@icbas.up.pt; 8Veterinary and Animal Research Centre (CECAV), Associate Laboratory for Animal and Veterinary Sciences (AL4AnimalS), University of Trás-os-Montes e Alto Douro, 5000-801 Vila Real, Portugal; 9ICBAS—Institute of Biomedical Sciences Abel Salazar, University of Porto, 4050-313 Porto, Portugal; 10Laboratório para a Investigação Integrativa e Translacional em Saúde Populacional (ITR), 4050-600 Porto, Portugal; 11Centre for the Research and Technology of Agro-Environmental and Biological Sciences (CITAB), University of Trás-os-Montes e Alto Douro, 5001-801 Vila Real, Portugal

**Keywords:** COVID-19, anti-SARS-CoV-2 antibodies, household, human–animal interaction

## Abstract

**Simple Summary:**

Domestic cats are highly susceptible to SARS-CoV-2. The close contact between humans and cats raises concerns about virus transmission from humans to cats. Thus, this study aims to investigate the presence of antibodies against SARS-CoV-2 in client-owned cats from Portugal, which is an indicator of exposure to the virus. A total of 176 cats, belonging to 94 households, were sampled, and cat owners answered an online questionnaire. Twenty households reported at least one confirmed human COVID-19 case. Forty cats belonged to COVID-19-positive and 136 to COVID-19-negative households. The percentages of cats exhibiting antibodies against SARS-CoV-2 from COVID-19-positive and -negative households were 5.0% and 0.7%, respectively. The two positive cats from COVID-19-positive households had an indoor lifestyle, and their owners maintained a close and frequent contact with them, even after being diagnosed with COVID-19, pointing towards human-to-cat transmission. The positive cat from the COVID-19-negative household had a mixed indoor/outdoor lifestyle and chronic diseases. Owners of the three positive cats did not notice clinical signs or behavior changes. This study highlights the low risk of SARS-CoV-2 transmission from humans to cats, even in a context of close and frequent contact.

**Abstract:**

The close contact between humans and domestic cats raises concerns about the potential risks of SARS-CoV-2 transmission. Thus, this study aims to investigate anti-SARS-CoV-2 seroprevalence in client-owned cats from Portugal and evaluate the infection risk of cats that maintain contact with human COVID-19 cases. A total of 176 cats, belonging to 94 households, were sampled. Cat owners answered an online questionnaire, and cats were screened for antibodies against SARS-CoV-2 using a commercial ELISA. Twenty (21.3%) households reported at least one confirmed human COVID-19 case. Forty cats (22.7%) belonged to a COVID-19-positive and 136 (77.3%) to a COVID-19-negative household. The seroprevalences of cats from COVID-19-positive and -negative households were 5.0% (2/40) and 0.7% (1/136). The two SARS-CoV-2-seropositive cats from COVID-19-positive households had an indoor lifestyle, and their owners stated that they maintained a close and frequent contact with them, even after being diagnosed with COVID-19, pointing towards human-to-cat transmission. The SARS-CoV-2-seropositive cat from the COVID-19-negative household had a mixed indoor/outdoor lifestyle and chronic diseases. Owners of the three SARS-CoV-2-seropositive cats did not notice clinical signs or behavior changes. This study highlights the low risk of SARS-CoV-2 transmission from COVID-19-positive human household members to domestic cats, even in a context of close and frequent human–animal contact.

## 1. Introduction

In December 2019, cases of pneumonia of unknown origin were reported in the city of Wuhan, located at the Chinese province of Hubei. A novel coronavirus (2019-nCoV) was identified as the causative agent of the disease by Chinese authorities [1], and the disease officially designated as Coronavirus Disease of 2019 (COVID-19) was classified as pandemic by the World Health Organization (WHO). Afterwards, due to the severe acute respiratory syndrome caused by the new coronavirus, the designation of the virus changed to Severe Acute Respiratory Syndrome Coronavirus 2 (SARS-CoV-2) [2]. 

SARS-CoV-2 belongs to the genus *Betacoronavirus* and is a spherical, enveloped virus with surface projections that give rise to the corona appearance (spike proteins). SARS-CoV-2 contains a large positive-sense RNA genome, which is wrapped up in helical nucleocapsid [3].

The first reports of COVID-19 human outbreaks were documented in Wuhan, attributed to the consumption of wild animals, traded in markets named “wet markets”. The WHO does not confirm specifically this source; however, the diversity of susceptible species and the biological and virological characteristics of SARS-CoV and SARS-CoV-2-related viruses, mainly based on spike protein plasticity, strongly suggests a propensity for these viruses to cross the species barrier, particularly in the context of frequent contact [4,5,6,7]. 

In the face of this, the close association between humans and companion animals raised concerns about the potential risks of SARS-CoV-2 transmission from humans to animals (“reverse zoonosis”) and about the possible role infected animals could play in perpetuating the spread of COVID-19 [2,5]. The role of domestic cats (*Felis catus*) has aroused particular interest, because they are one of the most popular companion animals and often establish close and frequent contact with humans [8,9]. Furthermore, cats are known to be infected by an *Alphacoronavirus*, the Feline coronavirus (FCoV), that is endemic in cat populations and responsible for a mild enteric disease [10,11]. However, FCoV can evolve through a process of “internal mutation” into a severe and mostly lethal form, responsible for feline infectious peritonitis (FIP) [12]. 

Furthermore, it was demonstrated that SARS-CoV-2 uses angiotensin-converting enzyme 2 (ACE2) as cellular receptor [7,13] and that feline ACE2 protein is closely related to human ACE2, sharing a high amino acid sequence identity (85.2%) [14]. Indeed, feline and human ACE2 differ in only four out of the twenty residues of the ACE2 conforming receptor binding pocket [15], justifying the high susceptibility of felids to SARS-CoV-2 infection, already confirmed through several experimental studies [16,17,18,19,20]. After experimental infection, cats may remain clinically asymptomatic [17] but be capable of transmitting SARS-CoV-2 to other cats [17,19] or may develop symptomatic infections, including severe disease and death. Histopathologic studies revealed massive lesions in the nasal and tracheal mucosa epitheliums and in the lungs [19].

Natural infection has been widely reported in cats after known or suspected close contact with human COVID-19 cases [21,22]. Transmission from humans to cats is suggested by sequence analysis of viral genomes isolated from domestic cats and humans, which has revealed a high degree of sequence conservation [23,24]. As in the case of experimental infection, some naturally infected cats remain asymptomatic [25], but others develop clinical disease, ranging from mild upper respiratory disease, including oral lesions and tongue ulceration, fever, sneezing, and ocular discharge, to moderate respiratory and/or gastrointestinal disease [26]. 

Serological investigations have revealed variable prevalence of infection in cats. In Wuhan, 14.7% of domestic cats, sampled from a local veterinary hospital during the outbreak, tested positive for anti-SARS-CoV-2 antibodies [27]. However, samples taken from cats in 2020 in Germany and the United States revealed much lower seroprevalences, ranging from 0.7% to 8% [28,29]. Higher seroprevalences were frequently reported in cats from households with human COVID-19 cases [22,30], highlighting the risk of human-to-animal SARS-CoV-2 transmission [31].

This observational retrospective study aims to investigate anti-SARS-CoV-2 seroprevalence in client-owned cats from Portugal and determine the risk of natural infection of domestic cats that maintained sustained contact with human household members. 

## 2. Materials and Methods 

### 2.1. Animal Recruiting and Sampling

Convenience sampling was used to select veterinary centers (clinics and hospitals) for this investigation, as we were living during a pandemic and facing long periods of confinement and conditional travel. Eighteen veterinary centers from mainland Portugal were invited by email to participate in this study. Veterinary centers (8/18) that agreed to collaborate received detailed instructions for sample collection and storage, informed consent form, and a link to receive access to an online questionnaire for owners. 

During healthcare visits, veterinary practitioners from collaborating veterinary centers invited cat owners to participate in the study. Cat owners who agreed to collaborate answered an online questionnaire designed to collect background information. Blood samples were collected according to veterinary norms into dry tubes and then centrifugated at 500 rcf for 10 min. Supernatants were transferred to 2 mL microtubes and stored at −20 °C until sent to Escola Superior Agrária de Viseu (ESAV) laboratory. Sample collection took place between November 2020 and October 2021.

### 2.2. Background Data Collection

A questionnaire was developed using an online platform (Google Forms^®^, Google LLC, Mountain View, California, United States) to collect data from each cat and household. The questionnaire was prepared in Portuguese language and consisted of 29 questions, of which 27 were closed-ended (dichotomic, multiple choice) and two were open-ended. The questionnaire covered five main topics, specifically: characterization of households, including human and animal elements and human–animal interaction (11 questions); characterization of sampled cats, including full signalment, lifestyle, prophylactic, and medical history (13 questions); and diagnoses of COVID-19 in cat owner and/or in other human members of the household (2 dichotomic questions, Yes/No). If respondents answered Yes to one of these two questions, they would answer questions on another topic designed to characterize COVID-19 diagnoses and interaction with the cat during the period of isolation (three questions) (Appendix B Table A1. Questionnaire with English translation). For internal validation, the questionnaire was evaluated by the authors.

### 2.3. Detection of Anti-SARS-CoV-2 Antibodies

Serum samples were screened for antibodies against the nucleocapsid (N protein) of SARS-CoV-2 using a commercial and already validated multi-species indirect ELISA (ID Screen^®^, ID.Vet, Grabels, France) [32,33,34,35]. The N protein recombinant antigen enables the detection of SARS-CoV-2-specific antibodies in sera, irrespective of their isotype. Testing was performed following the manufacturer’s instructions. Briefly, 25 µL of serum samples and positive and negative controls were diluted in dilution buffer (1:2) and analyzed in duplicate. Optical density (OD) was measured at a wavelength of 450 nm on a microplate reader MB 580 (Heales, Shenzhen Huisong Technology Development Co., Ltd., Shenzhen, China). For each sample, S/P% was calculated as follows: (ODSample-ODNegative Control)/(ODPositive Control-ODNegative Control) × 100, with serum samples presenting S/P% ≥ 60% being considered as positive, between 49% and 59% considered doubtful, and ≤50% considered negative. Reported sensitivity and specificity was 100% [35]. Internal validation from ID.Vet reported a specificity of 98.9% (*n* = 92) for cats. 

### 2.4. Data Processing and Statistical Analysis

Data collected from Google Forms^®^ and serologic analyses were downloaded into a database (Microsoft Excel 2016^®^; Microsoft Corp., Redmond, WA, USA). Statistical analysis was performed with SPSS v.27.0 (IBM Corp., Armonk, NY, USA, 2020). Descriptive statistics were used to analyze data. Odds ratio was calculated to evaluate the association between cat exposure to human COVID-19 cases and the presence of antibodies against SARS-CoV-2.

Homes where at least one person tested positive for SARS-CoV-2 were classified as COVID-19-positive households, and those without confirmed human cases of COVID-19 were classified as COVID-19-negative households.

### 2.5. Ethical Approval 

The questionnaire was approved by the ethics committee of the Instituto Politécnico de Viseu (IPV), Viseu, Portugal. Animal sampling was approved by the committee for Animal Welfare (ORBEA) of IPV. Written consent from each owner was collected after they were informed about the study.

## 3. Results

### 3.1. Geographic Distribution of the Sampled Cats

A total of 176 cats, belonging to 94 different households, were sampled. Serum samples were obtained from 10 (of the 18) districts of mainland Portugal, although most were collected in the districts of Porto (44.9%) and Braga (26.7%) in the North Region of Portugal. In total, 3 out of 176 cats (1.7%, 95% confidence interval: 0.35–4.9) tested positive for antibodies against SARS-CoV-2, one from Porto, another from Braga, and another from the district of Évora, located in the south, in Alentejo region. Most serum samples (65.0%) were obtained between June and August 2021, although the collection period was extended until the end of October 2021. Of the 10 districts investigated, there was a greater number of human COVID-19 cases in the districts of Porto and Braga at the end of the serum sample collection period (Figure 1). 

The number of sampled cats by household was variable, but in most of the households (79.8%) only one cat was screened. In 10.6%, 2.1%, 2.1%, and 1.1% of the households, two, three, four, and five cats were sampled, respectively. In four households, it was possible to collect blood from 10, 13, 16, and 21 cats during veterinary medical care provided at home. 

### 3.2. Characterization of Households 

Background information about 94 households was obtained using an online questionnaire, including cat owner sex, age, educational qualification, professional activity, and district of residence. Collected information also allowed the characterization of the household, including human members and pets (Table 1).

#### COVID-19-Positive Households

In total, 20 (21.3%) households reported at least one confirmed case of human COVID-19 and were classified as positive households. Specifically, fifteen cat owners (16.0%) were diagnosed with COVID-19, and in 18 (19.1%) households, other human members were diagnosed with COVID-19. According to owners, the diagnoses of human COVID-19 cases were performed by RT-qPCR from nasopharyngeal or oropharyngeal swabs (90.0%), blood collection for serology (5.0%), or rapid antigen test from nasopharyngeal or oropharyngeal swab (5.0%). Human COVID-19 diagnoses were carried out after January 2021 (55.0%), between October and December 2020 (35.0%), and between April and September 2020 (10.0%). 

COVID-19-positive households lived in Porto (45.0%), Aveiro (10.0%), Viseu (10.0%), Setúbal (10.0%), Évora (10.0%), Braga (5.0%), Beja (5.0%), and Faro (5.0%) districts. Of the 176 sampled cats, 136 (77.3%) belonged to COVID-19-negative households and 40 (22.7%) to COVID-19-positive households. The prevalence of seropositivity in cats from COVID-19-positive households was 5.0% (2/40) and from COVID-19-negative households was 0.7% (1/136), which correspond to an odds ratio of 7.2 (Figure 2). 

### 3.3. Characterization of Cats 

Background information on the 176 cats was obtained using an online questionnaire that allowed the full characterization of animals, including signalment, environment, clinical status (appearance of clinical signs or behavioral changes during COVID-19 pandemic), and human–animal interaction before pandemic and after the diagnoses of COVID-19 human cases in the household. According to their owners, only 22 cats had chronic diseases, and 9 of them received medication. Owners of 31 cats recognized behavioral changes (21/31) or clinical signs (10/31) in their cats during the COVID-19 pandemic (Table 2).

#### SARS-CoV-2-Seropositive Cats

ELISA testing of the three SARS-CoV-2-seropositive cats presented S/P% of 136.7343 (OD = 1.7203), 242.0433 (OD = 2.3851), and 145.144 (OD = 1.7498). S/P% mean ± standard deviation of seropositive cats was 174.6405 ± 58.5237 (OD = 1.9517 ± 0.3756), while S/P% of seronegative cats was 3.3105 ± 4.8963 (OD= 0.1589 ± 0.0539). No cats tested doubtful. OD mean ± standard deviation of negative controls was 0.1400 ±0.0369 and of positive controls was 1.2010 ± 0.0994 (Figure 3, Appendix A). 

SARS-CoV-2-seropositive cats were neutered or sterilized, had an indoor or mixed indoor/outdoor lifestyle, and cohabited with other animal species. Owners of Cat 1 and Cat 2 stated that they established close and frequent interaction with their cats and that after the diagnoses of human COVID-19 cases, household members maintained the same type and frequency of contact with cats. Owners did not notice clinical signs (respiratory, gastrointestinal, conjunctivitis, or others) or behavior changes. Cat 3 was infected with Feline Immunodeficiency Virus (FIV), had a basal cell carcinoma, and received corticosteroids for a long time (Table 3).

The owners of two SARS-CoV-2-seropositive cats (Cat 1 and Cat 2) and other human members of the households were diagnosed COVID-19-positive by RT-qPCR between October 2020 and January 2021 (it was not possible to determine the exact date). Four cats from the Cat 1 household were screened for antibodies against SARS-CoV-2, but none tested positive. Although there were other pets (dogs and another cat) in the Cat 2 household, only Cat 2 was tested. The COVID-19-negative household of Cat 3 had other pets (cats, dogs, and birds), but only Cat 3 was screened for anti-SARS-CoV-2 antibodies. The time interval between COVID-19 diagnoses in the household human members of Cat 1 and date of Cat 1 blood sampling was eight months. It was not possible to obtain the exact diagnose dates of COVID-19 human cases in Cat 2 and Cat 3 households. 

## 4. Discussion

A considerable number of studies indicated that domestic cats are highly susceptible to experimental and natural SARS-CoV-2 infection [17,19,21,22]. Cats are one of the most popular companion animals and often establish close contact with humans. Although variable, the interaction between domestic cats and humans can be as close as sleeping in the same bed and licking the owner’s face [9]. 

There are approximately 1.4 million domestic cats in Portuguese homes [36] that are kept, in most cases, permanently indoor [8]. Most Portuguese owners establish daily contact with their cats, such as playing, petting, and grooming [37]. Therefore, the susceptibility of domestic cats to SARS-CoV-2 under natural conditions must be investigated to make reasoned recommendations regarding the management of cats in SARS-CoV-2-positive households.

Previous studies have found a low prevalence of SARS-CoV-2-positive cats by RT-qPCR [38,39,40], but serological studies have revealed higher seroprevalences [41,42]. Thus, serological tests seem to be preferred when the objective is to determine SARS-CoV-2 exposure in companion animals. However, as cats can be infected by FCoV, another coronavirus genetically distinct from SARS-CoV-2, the possibility of cross-reactions has been raised [42]. Some studies employing indirect ELISA assay based on the receptor-binding domain (RBD) of SARS-CoV-2 [27,29], along with a multiplex microsphere immunoassay based on nucleoprotein and spike (S) subunit 1 and 2 [31], have already ruled out serological cross-reactivity of FCoV-specific antibodies with SARS-CoV-2 [27,29,31], reinforcing the usefulness of serological methods to identify exposure to SARS-CoV-2. Furthermore, a study using the same commercial ELISA assay based on N protein that was employed in the present study reported a sensitivity and specificity of 100% [35], and the manufacturer internal validation report reported a specificity of 98.9%, confirming the ability of the ID Screen^®^ ELISA assay to discriminate between anti-SARS-CoV-2 and anti-FCoV antibodies.

In this study, we determined anti-SARS-CoV-2 seroprevalence in a sample of 176 cats belonging to 94 households of mainland Portugal, although most samples were obtained in the districts of Porto and Braga in the North Region. The restrictions imposed by the pandemic concerning the movement of people and the functioning of veterinary centers may justify the poor adherence of veterinary centers to the study and the need to extend the serum sample collection period. On the other hand, the uncertainty about the role of companion animals in the spread of the virus and the news of euthanasia and abandonment of thousands of pets [43] may have discouraged veterinary practitioners from approaching the topic and inviting cat owners to participate in the study. However, the collection of background information about cats and human household members allowed the complete characterization of households, namely the human–animal interaction and the COVID-19 status of the households, which represents an advantage of the present study.

Published serological studies have shown distinct seropositivities in companion animals related to human infection [28,29,40,44]. In our study, performed on a sample of cats presented for routine veterinary consultation, the overall seroprevalence was 1.7%. Serum sample collection took place between November 2020 and October 2021, although most samples were collected between June and August 2021. At the end of October 2021, when the collection of serum samples ended, the cumulative number of cases of human COVID-19 in Portugal was 1.09 million. Seropositivity found in this study is slightly higher than the seroprevalence (0.69%) found in the first large-scale survey conducted in Germany from April to September 2020, a period when the incidence of human COVID-19 in the country was still rather low [29], and also higher than the seroprevalence (0.36%) obtained in a large-scale survey conducted in Thailand from April to December 2020 [35]. The low seroprevalences observed in some studies, including ours, may be related to the prevalence of SARS-CoV-2 human infection registered before cat sample collection. Indeed, the time interval between SARS-CoV-2 cat infection and sample collection is decisive, since while variable, antibody persistence seems to not be lasting [40,41,45], as will be discussed later. 

On the other hand, the characteristics of the immune response developed by some SARS-CoV-2-infected cats may, in part, justify low seroprevalences observed. The absence of seroconversion was already demonstrated for humans [46] and cats [33]. As suggested for SARS-CoV-2-infected humans, the absence of seroconversion may occur in some relatively mild infections, where the immune response is restricted to the respiratory tract mucosal cells and is dominated by the secretion of IgA with limited IgG production. On the other hand, exposure to SARS-CoV-2 may induce T-cell-mediated specific immune response, without activation of B-cells, as suggested for humans [46,47].

However, the seroprevalence obtained in the present work was much lower compared with the seroprevalence obtained in another Portuguese study (21.7%) [41], which collected samples from 69 cats between December 2020 and May 2021, mainly in the district of Braga (56 cats). While samples in that study were collected in veterinary centers (hospitals, clinics) and shelters, in our study, all sera samples were obtained in veterinary centers. However, apparently, the most appreciable difference between the two studies was the ELISA assay employed. While samples in our study were screened using a commercial ELISA based on SARS-CoV-2 N protein, Barroso and colleagues used an in-house ELISA assay based on RBD of SARS-CoV-2. The sensitivities of the two ELISA assays may be different due to specific antibody responses raised against the viral N protein and directed against the RBD of S antigens. Indeed, a population-based study showed that antibody responses against viral S and N proteins are equally sensitive in the acute phase of infection, but while response against N decreases in the post-infection phase, the response against the S protein persists [48]. Thus, it is probable that our study underestimate seroprevalence, as previously reported [31,34].

This study included 94 households from 10 out 12 districts of mainland Portugal. A higher number of households from the districts of Braga, Porto, and Évora were investigated. Of the 10 districts surveyed, there was a greater number of human COVID-19 cases in the districts of Porto and Braga at the end of the serum sample collection period. Not surprisingly, the three SARS-CoV-2-seropositive cats were identified in these three districts.

Considering COVID-19 status, our study included 20 negative and 76 positive households. Of the three seropositive cats identified in this study, two belonged to COVID-19-positive households, and one cat lived in a COVID-19-negative household. Thus, the seropositivity observed in cats from COVID-19-positive households was 5.0% and in cats from COVID-19-negative households was 0.7%. Several studies have already reported the infection of cats following natural exposure to infected people [27,49]. In line with this, high seropositivities were observed in cats living in COVID-19-positive households [30,31,38]. Recently, a longitudinal study, including 39 pets (29 dogs and 10 cats) living with human COVID-19 patients, identified close human–animal contact, such as sharing a bed with an owner with COVID-19, as the main risk factor for infection of the co-inhabiting animals [49]. As such, we hypothesize that the most likely source of infection of the two seropositive cats from COVID-19-positive households identified in this study was the contact with human infected members (“reverse zoonosis”). Indeed, both cats had an exclusively indoor lifestyle, which limits the possibility of infection in another environment, and according to owners, animals established close and frequent contact with humans, increasing the opportunities for virus spillover and infection. 

However, the source of infection of the cat belonging to the COVID-19-negative household is more difficult to identify, although some hypotheses may be raised. As this cat had a mixed indoor/outdoor lifestyle, there are the possibilities of infection via SARS-CoV-2 environmental contamination or cat-to-cat transmission. Reports of SARS-CoV-2-seropositive stray cats demonstrate the role of cat-to-cat transmission and/or environmental contamination as potential sources of SARS-CoV-2 infection [50,51,52]. Furthermore, cat-to-cat transmission in natural conditions is consistent with experimental studies that proved virus transmission from infected to naive cats via the airborne route [16,17,18,19]. Contact with COVID-19 asymptomatic human household members or other COVID-19 human patients is also a possibility. Meanwhile, other studies had already reported the presence of seropositive cats living in COVID-19-negative households [45], including those without outdoor access [16,41]. 

Determination of antibody longevity is crucial to evaluate humoral response to SARS-CoV-2 under natural conditions. Experimental studies determined that cats seroconverted 7 to 14 days post-infection [19]; however, the duration of antibodies is variable. One cat tested twice three months apart turned seronegative in the second test, suggesting that longevity of anti-SARS-CoV-2 antibodies may not be lasting [41]. In the present study, one cat tested positive eight months after the diagnoses of COVID-19 in human household members, suggesting longer antibody longevity. Antibodies against SARS-CoV-2 were identified in an Italian cat living in a COVID-19-affected household six months after being tested positive by RT-qPCR [40], indicating the persistence of humoral immunity for long time. 

One of three seropositive cats identified in this study was co-infected with FIV. Viral co-infection was already reported in other studies [41,51] and may play an important role in the susceptibility and clinical outcomes of SARS-CoV-2-infected cats [41]. However, in this study, all SARS-CoV-2-seropositive cats, including the FIV-co-infected, were apparently asymptomatic, as their owners did not notice any clinical sign or behavioral change. This finding contrast with some studies where the percentage of symptomatic cats was high, such as the mortality in [41]. Probably, the nonspecific, mild, and reversible character of the clinical signs associated with COVID-19 in companion animals [49] prevented the recognition of the disease by cat owners. Thus, cats may act as silent hosts of SARS-CoV-2, as they may not show appreciable clinical signs, as previously reported by other authors [18,33,45]. 

Non-infectious comorbidities such as hypertension, heart diseases, obesity, chronic kidney and respiratory diseases, diabetes mellitus, and cancer were shown to increase the risk of the development of severe disease in humans with COVID-19 [53,54,55]. The presence of non-infectious comorbidities, such as intestinal B-cell lymphoma [56] and hypertrophic cardiomyopathy [57,58], was also reported in SARS-CoV-2-infected cats, and it was associated with moderate, severe, or even fatal disease. Despite the clinical condition of Cat 3, in whom the presence of FIV infection was associated with a neoplasm (basal cell carcinoma), the owners did not notice the presence of clinical signs. It can be speculated that the chronic administration of corticosteroids to this cat may have contributed to decreasing the inflammation associated with SARS-CoV-2 infection and the manifestation of clinical signs, although the beneficial effect of this drug in humans with COVID-19 is controversial [59].

Furthermore, SARS-CoV-2 variants may affect viral transmissibility, virulence, and consequently the severity and lethality of the infection, as occurs in humans [60,61]. At least five SARS-CoV-2 variants of concerns (VOC) have been classified by WHO, including the Alpha variant (B.1.1.7), Beta variant (B.1.351), Gamma variant (P.1), Delta variant (B.1.617.2), and Omicron (B.1.1.529). Alpha variant was isolated from cats with clinical conditions of varying severity, from asymptomatic [62], mild disease, characterized by sneezing [63], to severe conditions characterized by dyspnea and fever [64] or acute myocarditis [65]. The SARS-CoV-2 Gamma variant was identified as a cause of death of a FeLV-positive cat presenting severe acute respiratory syndrome and lesions in several organs [66]. The Delta variant was isolated from three cats from Harbin [67] and from a cat with history of anorexia, lethargy, soft stools, and vomiting [68]. Although it would be interesting to investigate the clinical picture exhibited by cats infected by different SARS-CoV-2 variants, the small number of viral genomes isolated and sequenced so far does not allow the establishment of an association between disease severity and SARS-CoV-2 variant in this species. Furthermore, we were not able to obtain information about SARS-CoV-2 infecting variant of COVID-19 human members.

Despite the apparent low risk of infection of cats living in households diagnosed with COVID-19, it is advisable to follow the recommendation of the European Advisory Board on Cat Diseases, namely keeping the cats indoors during quarantine and performing human–animal interactions under basic hygienic measures.

## 5. Conclusions

This study highlights the low risk of SARS-CoV-2 transmission from COVID-19-positive human household members to domestic cats, even in a context of close and frequent human–animal contact. 

## Figures and Tables

**Figure 1 vetsci-09-00363-f001:**
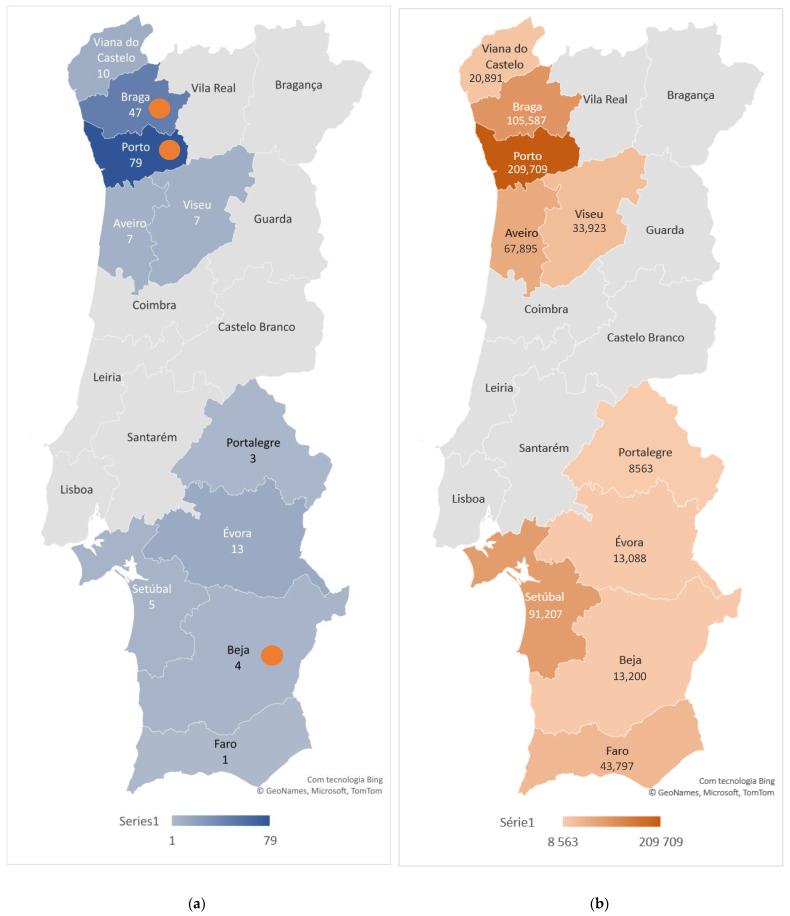
Geographical distribution of sampled cats and cumulative human COVID-19 cases. (**a**) The number of sampled cats by district is shown by the color gradient, as indicated in the legend. The district of residence of SARS-CoV-2-positive cats is marked with a circle. (**b**) The number of cumulative human COVID-19 cases at the end of October 2021 by district is shown by color gradient, as indicated in the legend.

**Figure 2 vetsci-09-00363-f002:**
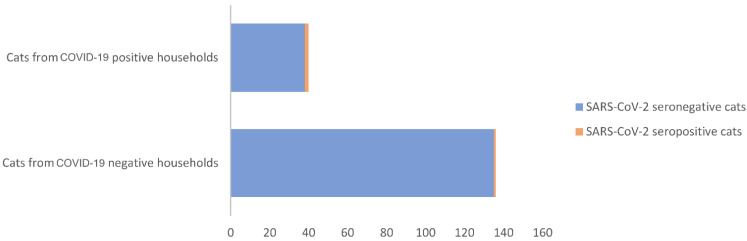
SARS-CoV-2 seropositivity among cats from COVID-19-positive and -negative households.

**Figure 3 vetsci-09-00363-f003:**
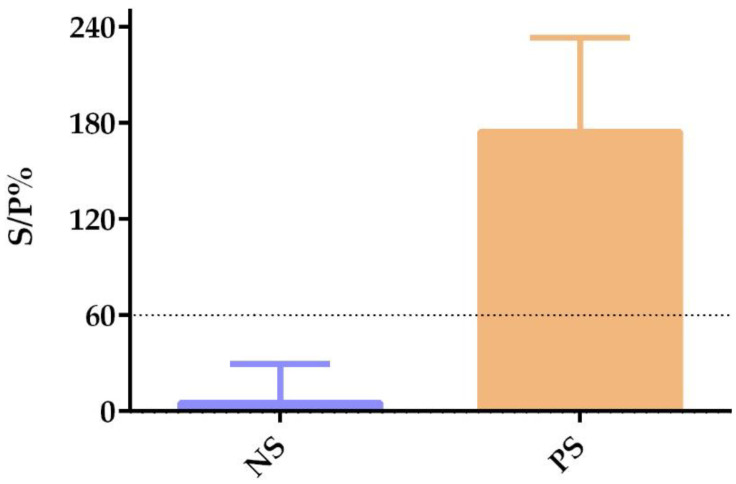
anti-SARS-CoV-2 antibody screening results by ELISA. S/P% of negative samples (NS) and positive samples (PS) are represented by bar graph showing mean + standard deviation. The dotted line represents S/P% cutoff.

**Table 1 vetsci-09-00363-t001:** Characterization of household, including demographics and household composition (*n* = 94; otherwise, *n* value is presented).

Category	Characteristics		Percentage (%)
Demographics	Sex	Female Male	86.2 13.8
	Age (years)	18–30 30–50 50–70 >70	21.3 43.6 29.8 5.3
	Educational qualification	Basic education High school Degree Master’s Other	20.2 26.6 28.7 17.0 7.5
	Professional activity	Employed Retired Unemployed Student	73.4 14.9 7.4 4.3
	District of residence	Porto Évora Braga Viseu Aveiro Beja Setúbal Portalegre Viana do Castelo Faro	50 13.8 12.8 5.3 5.3 4.2 3.2 3.2 1.1 1.1
Household	Human members	1 2 3 4 Other	14.9 38.3 21.3 21.3 4.2
	Pets (*n* = 79)	Dog Cat Bird	58.2 38.0 3.8

**Table 2 vetsci-09-00363-t002:** Characterization of sampled cats. *n* = 176; otherwise, *n* value is presented.

Category	Characteristics		Percentage (%)
Signalment	Sex	Female Male	52.8 47.2
	Reproductive status	Fertile Non-fertile	6.8 93.2
	Age (years)	1–5 5–10 >10	45.5 39.2 15.3
	Breed	European shorthair Siamese Persian Norwegian Forest cat Other	84.7 5.7 4.5 0.6 4.5
Environment	Lifestyle	Indoor Indoor/outdoor Outdoor	77.8 20.5 1.7
	Other animals	Yes No	91.5 8.5
	Co-habitant pets (*n* = 161)	Dog, cat Cat Dog Other	48.4 30.4 9.3 11.9
Clinical status	Annual vaccination	Yes No	78.4 21.6
	Deworming	Yes, every 3 months Yes, every 6 months Yes, annually Occasionally No	68.8 15.9 8.5 4.5 2.3
	Chronic disease	No Yes Don’t know	80.1 12.5 7.4
	Type of chronic disease (*n* = 22)	Kidney/liver disease Retroviral infection Heart disease Gingivostomatitis Hypothyroidism Other	22.7 22.7 18.2 9.1 4.6 22.7
	Medication (*n* = 9)	Food supplements (oral hygiene, feline idiopathic cystitis, behavior, immunomodulators) Non-steroidal anti-inflammatory drugs Steroidal anti-inflammatory drugs	66.7 22.2 11.1
	Clinical signs or behavior changes during pandemic	No Yes	82.4 17.6
	Type of clinical signs or behavior changes during pandemic (*n* = 31)	Behavioral changes Weight loss Respiratory clinical signs Appetite loss Digestive clinical signs Conjunctivitis Fever Skin problems Other	67.7 9.7 9.7 6.5 6.6 6.5 3.2 3.2 6.5
Interaction with humans	Before pandemic, type of interaction	Playing, petting, and cuddling Resting on the lap Sharing the bed Sharing the sofa Other	96.0 72.2 65.9 64.8 13.7
	Before pandemic, frequency of interaction	All day Part of the day Occasionally Did not interact	48.3 46.0 6.25 0.57
	After household human COVID-19 diagnoses (*n* = 36)	As usual Reduced interaction Did not interact	86.1 11.1 2.8

**Table 3 vetsci-09-00363-t003:** Characterization of SARS-CoV-2-seropositive cats.

Category	Characteristics	Cat 1	Cat 2	Cat 3
Signalment	Sex	Female	Male	Male
	Reproductive status	Neutered	Spayed	Spayed
	Age (years)	1–5	<1	>10
	Breed	European shorthair	European shorthair	Siamese
Environment	Lifestyle	Indoor	Indoor	Indoor/outdoor
	Co-habitants	Dog, cat	Dog, cat	Dog, cat, bird
Clinical status	Vaccination	Yes, annually	Yes, annually	No
	Deworming	Yes, every 3 months	Yes, every 3 months	Occasionally
	Chronic disease/treatment	No	No	Yes, FIV, basal cell carcinoma/corticosteroids
	Clinical signs or behavior changes during the pandemic	No	No	No
Interaction with humans/Frequency of interaction	Before the pandemic	Playing, petting, and cuddling Resting on the lap Sharing the bed/All day	Playing, petting, and cuddling Resting on the lap Sharing the bed/All day	Playing, petting, and cuddling Resting on the lap/Part of the day
	After COVID-19 diagnoses	As usual	As usual	-

## Data Availability

The data presented in this study are available in Appendix A.

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
