# Peer review of "Seroprevalence of SARS-CoV-2 in Client-Owned Cats from Portugal"

_vetsci, 2022, doi:10.3390/vetsci9070363_

Round 1
Reviewer 1 Report
This study presents the anti-SARS-CoV-2 IgG seroprevalence in client-owned cats and potential risk of SARS-Cov-2 transmission between humans and cats in Portugal, the results are interesting. However, there are several minors have to be addressed before publication:
1. The anti-SARS-CoV-2 IgG seroprevalence obtained in this study is much lower than previous estimations (some of which is greater than 40%). Is this possibly a biased estimation? The power of this study design needs a thorough discussion, especially the sensitivity of the antibody detection method employed.
2. The association between risk factors and SARS-CoV-2 infection in cats was mostly established on speculation. Are there any possible amendments to be made to increase the evidence about that.
3. Table 1 shows very limited information and can be deleted.
4. It will be better to use summary tables showing the data on the factors related to Households and cats.
A 95% confidence interval should be estimated for the seroprevalence.
Author Response
Reviewer 1
This study presents the anti-SARS-CoV-2 IgG seroprevalence in client-owned cats and potential risk of SARS-Cov-2 transmission between humans and cats in Portugal, the results are interesting. However, there are several minors have to be addressed before publication:
The authors are very grateful to Reviewer 1 for the constructive comments and suggestions raised.
- The anti-SARS-CoV-2 IgG seroprevalence obtained in this study is much lower than previous estimations (some of which is greater than 40%). Is this possibly a biased estimation? The power of this study design needs a thorough discussion, especially the sensitivity of the antibody detection method employed.
The text has been extensively restructured to include information on the sensitivity and specificity of the ELISA used, on the materials and methods (lines 158-160) and Discussion (lines 417-422; 469-480).
- The association between risk factors and SARS-CoV-2 infection in cats was mostly established on speculation. Are there any possible amendments to be made to increase the evidence about that.
We agree with reviewer 1, and due to the small sample size, we decided to eliminate the risk analysis. Odds ratio was eliminated from abstract and discussion.
- Table 1 shows very limited information and can be deleted.
Table 1 was removed, and the text has been slightly changed to include the information present in the table.
- It will be better to use summary tables showing the data on the factors related to Households and cats.
Two tables were created to summarize general information about households and sampled cats.
A 95% confidence interval should be estimated for the seroprevalence.
95% confidence interval was calculated (line 183).
Reviewer 2 Report
The manuscript is clear, well written, and data obtained are well presented and relevant for the field. Although there are previous articles showing evidence of SARSCoV-2 infection in domestic animals, including cats, this manuscript adds information to this issue, helping scientists to understand the importance of these animals in the spread and maintainence of SARSCoV-2 in the environment.
Results, Page 5, Figure 1. I suggest the authors to substitute this for a complete map of Portugal, as this Figure was a bit confusing.
Discussion Line 315. Please bring information on the types of ELISAs performed by different studies, as well as the sensitivity and specificity of the ELISAs used in this and in the previous studies, specially the one described by Barroso and Vieira-Pires, as the seroprevalences found are quite different. Information on timing of blood collection after the owners had COVID-19 should also be added, if available. Line 366. Discuss the possible different effects of SARSCoV-2 variants in the outcome of animal infection, as virus pathogenicity may vary according to the variant.
Author Response
Reviewer 2
The manuscript is clear, well written, and data obtained are well presented and relevant for the field. Although there are previous articles showing evidence of SARSCoV-2 infection in domestic animals, including cats, this manuscript adds information to this issue, helping scientists to understand the importance of these animals in the spread and maintainence of SARSCoV-2 in the environment.
Review on “Manuscript Seroprevalence of SARS-CoV-2 in client-owned cats from Portugal” The manuscript aims at the detection of anti-SARSCoV-2 antibodies in domestic cats, adding to the knowledge on the circulation of this virus among this species. This is an important issue to be addressed, as virus replication in other hosts may facilitate the occurrence of variants and such viruses may display different pathogenicity/host tropism.
General comments
The manuscript is clear, well written, and data obtained are well presented and relevant for the field. Although there are previous articles showing evidence of SARSCoV-2 infection in domestic animals, including cats, this manuscript adds information to this issue, helping scientists to understand the importance of these animals in the spread of SARSCoV-2.
We very much acknowledge the positive appreciation of our manuscript done by Reviewer 2.
Specific comments
Results, Page 5, Figure 1. I suggest the authors to substitute this for a complete map of Portugal, as this Figure was a bit confusing.
Figure 1 was changed to include a complete map of Portugal.
Discussion Line 315. Please bring information on the types of ELISAs performed by different studies, as well as the sensitivity and specificity of the ELISAs used in this and in the previous studies, specially the one described by Barroso and Vieira-Pires, as the seroprevalences found are quite different.
The text has been extensively restructured to include information on the types of ELISAs performed by different studies (lines 413-415) and on the sensitivity and specificity of the ELISA used in the present study (materials and methods, lines 158-160 and Discussion, lines 417-422 and 469-480). ELISA employed by Barroso and Vieira-Pires was an in-house ELISA and authors did not present sensitivity and specificity values. However, we elaborated about this on discussion (lines 468-480).
Information on timing of blood collection after the owners had COVID-19 should also be added, if available
Despite all our efforts, we were able to trace back to the month/year and not the exact date. Nevertheless, we have now introduced that information on the text (lines 246-247 and 371-380).
Line 366. Discuss the possible different effects of SARSCoV-2 variants in the outcome of animal infection, as virus pathogenicity may vary according to the variant
An exhaustive review of the isolated SARS-CoV-2 variants in cats was performed and discussed (lines 479-494).
Reviewer 3 Report
This article written by Oliveira and others at conducts a seroprevalence study of client-owned domestic cats in Portugal and investigates the environment the cats lived in and how that could have affected transmission. Through an ELISA and an online questionnaire, the authors determine positivity rates and learn about the environment the cats were in and if they were seen displaying clinical signs by the owners. The authors have found that the risk of transmission from humans to their pet cats is low, and the animals also did not display clinical signs of disease as perceived by their owners. This article adds important scientific knowledge about seroprevalence and pet owner behavior as it relates to cat infection with SARS-CoV-2, in Portugal and in general.
This article fits the scope of the Veterinary Sciences journal as it relates to veterinary medicine, epidemiology, immunology, microbiology, virology, and the concept of "One Health" since the context of the cat infection revolves around whether their human owners were infected. Importantly, the aims set out by this group of authors is very relevant to the management of cats by veterinary practitioners that could also be extrapolated to management of other felid species (in the wild and captivity).
The article is generally well-written and presented, is easy to read and understand, and contains relevant and necessary background information in both the Introduction and Discussion sections.
However, major revisions are needed in the Materials and Methods, Results, and Discussion sections, primarily in terms of clarification, adding relevant methodology, figures, and data, and discussions related to the findings. These are enumerated below:
Materials and Methods
1. Add more information on the ELISA assay used. This is the central assay in this paper and needs to be further explained. When one looks at the website for the assay, there is not much information provided so this needs to be added to this article methodology.
- Brief methodology: how many replicates of each cat sample were tested, volume of cat sample used, sample dilution (if done), controls used (both positive and negative), how the OD cutoff was determined for positive versus negative result
- Information on specificity and sensitivity, especially since this is brought up later in the discussion
Results
1. Add a figure/graph for the OD values/results, including: values for positive and negative cats, positive and negative controls, and OD cutoff for positivity/negativity
2. Include raw data for all cat sample OD values (either as Supplementary/Appendix material in this submission or in an online database)
3. It is unclear whether all the other cats living in the three households with the positive cats were tested and what those results were. Please add this information in the text or in Table 2: how many other cats lived in the same household and how many of those extra cats were also tested for seropositivity, and their results.
Discussion
1. Serological cross-reactivity of SARS-CoV-2 to other CoVs (line 285): the authors mention that other groups have looked at this cross-reactivity. Was this also the case for the assay used in this study? Please mention this or elaborate in the text.
2. Comparison to other published serological studies (line 301): the authors compare to other studies and discuss generally the results from this study and others, and conclude that the studies were different and that multiple variables could have led to these differences (sampling method, period of collection, assay sensitivity/specificity). Please elaborate on these (how were the sampling methods different, what were the sensitivities/specificities for those assays and for the assay used in this study) and how they could account for the differences.
3. Co-morbidities in one of the positive cats and how that could affect results/disease (line 360): the fact that this cat also had FIV is nicely discussed. What about the other co-morbidities and drugs this cat was on, would those have an effect or have others looked at whether there is an effect from those conditions? Steroids have also been used for human COVID-19 treatment. Please mention/discuss (can be brief).
4. SARS-CoV-2 variants and possible effect on cat clinical disease and reverse zoonotic transmission: the authors comment on the difference in clinical symptoms and seropositivity in this study compared to others (line 365). Humans show difference in clinical symptoms and transmission rates with different SARS-CoV-2 variants. Has this been investigated in cats? Do the authors have information on what variants were detected in the pet owners of the positive cats in this study, or is it known what was the predominant variant circulating in Portugal at the time of sampling/infection of the pet owners? Conducting studies to determine this answer (if this information is not known) is outside of the scope of this paper, but whether others have looked at this question and if different clinical symptoms or increased/decreased transmission to cats have been found would be worth mentioning (briefly). A Science article by David Grimm published on 19 MAR 2021 brings up this question and some findings in cats that had a VOC detected.
Overall, this article poses important aims and the findings support the conclusions. However, more information is needed to verify the claims put forward by the authors.
Author Response
Reviewer 3
This article written by Oliveira and others at conducts a seroprevalence study of client-owned domestic cats in Portugal and investigates the environment the cats lived in and how that could have affected transmission. Through an ELISA and an online questionnaire, the authors determine positivity rates and learn about the environment the cats were in and if they were seen displaying clinical signs by the owners. The authors have found that the risk of transmission from humans to their pet cats is low, and the animals also did not display clinical signs of disease as perceived by their owners. This article adds important scientific knowledge about seroprevalence and pet owner behavior as it relates to cat infection with SARS-CoV-2, in Portugal and in general.
This article fits the scope of the Veterinary Sciences journal as it relates to veterinary medicine, epidemiology, immunology, microbiology, virology, and the concept of "One Health" since the context of the cat infection revolves around whether their human owners were infected. Importantly, the aims set out by this group of authors is very relevant to the management of cats by veterinary practitioners that could also be extrapolated to management of other felid species (in the wild and captivity).
The article is generally well-written and presented, is easy to read and understand, and contains relevant and necessary background information in both the Introduction and Discussion sections.
However, major revisions are needed in the Materials and Methods, Results, and Discussion sections, primarily in terms of clarification, adding relevant methodology, figures, and data, and discussions related to the findings. These are enumerated below:
The authors are very grateful to Reviewer 3 for the constructive comments and suggestions raised. The questions raised substantially enriched the text, especially the discussion section.
Materials and Methods
- Add more information on the ELISA assay used.This is the central assay in this paper and needs to be further explained. When one looks at the website for the assay, there is not much information provided so this needs to be added to this article methodology.
- Brief methodology: how many replicates of each cat sample were tested, volume of cat sample used, sample dilution (if done), controls used (both positive and negative), how the OD cutoff was determined for positive versus negative result
As recommended, methodology was briefly described (lines 143-152).
- Information on specificity and sensitivity, especially since this is brought up later in the discussion
The text has been extensively restructured to include information on the sensitivity and specificity of the ELISA used in the study, both in materials and methods (lines 152-154) and Discussion (lines 418-423).
Results
- Add a figure/graph for the OD values/results, including: values for positive and negative cats, positive and negative controls, and OD cutoff for positivity/negativity
Figure 3 was added containing S/P% values of positive and negative samples and cutoff. As S/P% calculation depends on optical density of controls, is not possible to include S/P% of controls in the graph. However, mean and standard deviation of optical density and S/P% of controls (positive and negatives), and samples was described in the results section (lines 341-346).
- Include raw data for all cat sample OD values(either as Supplementary/Appendix material in this submission or in an online database)
Cat samples OD values will be included in an online database as suggested.
- It is unclear whether all the other cats living in the three households with the positive cats were tested and what those results were.Please add this information in the text or in Table 2: how many other cats lived in the same household and how many of those extra cats were also tested for seropositivity, and their results.
The text has been changed by introducing all the information requested (lines 368-377).
Discussion
- Serological cross-reactivity of SARS-CoV-2 to other CoVs (line 285):the authors mention that other groups have looked at this cross-reactivity. Was this also the case for the assay used in this study? Please mention this or elaborate in the text.
The information requested was included in the text (lines 414-416; 418-423).
- Comparison to other published serological studies (line 301): the authors compare to other studies and discuss generally the results from this study and others, and conclude that the studies were different and that multiple variables could have led to these differences (sampling method, period of collection, assay sensitivity/specificity). Please elaborate on these (how were the sampling methods different, what were the sensitivities/specificities for those assays and for the assay used in this study) and how they could account for the differences.
The text has been substantially improved in order to answer this question, comparing our study with others with low (lines 446-451) and high seroprevalencies (lines 470-481).
- Co-morbidities in one of the positive cats and how that could affect results/disease (line 360): the fact that this cat also had FIV is nicely discussed. What about the other co-morbidities and drugs this cat was on, would those have an effect or have others looked at whether there is an effect from those conditions? Steroids have also been used for human COVID-19 treatment. Please mention/discuss (can be brief).
The topic was discussed (lines 567-578).
- SARS-CoV-2 variants and possible effect on cat clinical disease and reverse zoonotic transmission: the authors comment on the difference in clinical symptoms and seropositivity in this study compared to others (line 365). Humans show difference in clinical symptoms and transmission rates with different SARS-CoV-2 variants. Has this been investigated in cats? Do the authors have information on what variants were detected in the pet owners of the positive cats in this study, or is it known what was the predominant variant circulating in Portugal at the time of sampling/infection of the pet owners? Conducting studies to determine this answer (if this information is not known) is outside of the scope of this paper, but whether others have looked at this question and if different clinical symptoms or increased/decreased transmission to cats have been found would be worth mentioning (briefly). A Science article by David Grimm published on 19 MAR 2021 brings up this question and some findings in cats that had a VOC detected.
An exhaustive review of the isolated SARS-CoV-2 variants in cats was performed and discussed (lines 579-594).
Overall, this article poses important aims and the findings support the conclusions. However, more information is needed to verify the claims put forward by the authors.
Round 2
Reviewer 3 Report
Thank you for updating with requested information. I have no further scientific comments that need to be addressed.
Author Response
Dear reviewer
The corrections were introduced.
Best regards